# Reversible Power-to-Gas systems for energy conversion and storage

Gunther Glenk [1]✉ & Stefan Reichelstein [2]

In the transition to decarbonized energy systems, Power-to-Gas (PtG) processes have the potential to connect the existing markets for electricity and hydrogen. Specifically, reversible PtG systems can convert electricity to hydrogen at times of ample power supply, yet they can also operate in the reverse direction to deliver electricity during times when power is relatively scarce. Here we develop a model for determining when reversible PtG systems are economically viable. We apply the model to the current market environment in both Germany and Texas and find that the reversibility feature of unitized regenerative fuel cells (solid oxide) makes them already cost-competitive at current hydrogen prices, provided the fluctuations in electricity prices are as pronounced as currently observed in Texas. We further project that, due to their inherent flexibility, reversible PtG systems would remain economically viable at substantially lower hydrogen prices in the future, provided recent technological trends continue over the coming decade.

[1] Mannheim Institute for Sustainable Energy Studies, University of Mannheim, MIT CEEPR, Massachusetts Institute of Technology, Cambridge, MA, USA.
[2] Mannheim Institute for Sustainable Energy Studies, University of Mannheim, Graduate School of Business, Stanford University, Leibniz Centre for European Economic Research (ZEW), Mannheim, Germany. ✉email: glenk@uni-mannheim.de

The large-scale deployment of intermittent renewable energy sources, like wind and solar, poses a growing challenge in terms of balancing energy demand and supply in real time[1,2]. Aside from storage in batteries[3,4], electrolytic hydrogen production via Power-to-Gas (PtG) processes can absorb electricity during times of ample power supply and thereby yield hydrogen for industrial customers[5–7]. Conversely, PtG systems that are also capable of operating in the reverse direction can convert hydrogen back to electricity during periods of limited power supply and correspondingly high power prices[8,9]. Thus, reversible PtG systems can effectively connect the markets for hydrogen and electricity[10–12] and, in the process, limit the growing price volatility in electricity markets[13,14].

Reversible PtG systems can be designed in a modular manner, for instance, by combining a one-directional electrolyzer for hydrogen production with a one-directional fuel cell or gas turbine for power generation[15,16]. While electrolyzers have been found to become increasingly cost-competitive in producing hydrogen[17,18], fuel cells and gas turbines have so far been regarded as too expensive for converting hydrogen back to electricity that would subsequently be sold in wholesale markets[9,19,20]. In contrast, unitized regenerative fuel cells, which we refer to as integrated PtG systems, utilize the same equipment to deliver either hydrogen or electricity depending on the prevailing electricity prices at different points in time[21–23].

This paper first develops an analytical model of the unit economics of reversible PtG systems. Our findings show that the technological characteristics of both modular and integrated systems entail certain ranges for hydrogen prices at which reversible PtG systems become cost-competitive. While modular systems require sufficiently low hydrogen prices in order for the reversibility feature to be valuable, integrated systems can be economically viable for higher hydrogen prices by primarily generating hydrogen but also providing electricity during times of limited power supply. Such operations will therefore not only increase the supply of hydrogen but also provide an effective buffer against the intermittency of renewable power sources.

The empirical part of our analysis calibrates the model in the context of the electricity markets in Germany and Texas. Despite improvements in the cost and conversion efficiency of modular PtG systems[24,25], we confirm the findings of earlier studies that there is no economic case, either now or in the foreseeable future, for investing in modular systems that convert hydrogen back to electricity. In contrast, integrated PtG systems based on solid oxide cell (SOC) technology are shown to be competitive at current hydrogen prices, given sufficient variation in daily electricity prices, as is already encountered in the Texas market. For such systems, it is indeed efficient to mostly produce hydrogen and respond to sufficiently high electricity prices with electric power production. Owing to their relatively high capacity utilization, integrated systems are also positioned more competitively than one-directional electrolyzers on their own.

Finally, we project that if recent trends regarding the acquisition cost and conversion efficiency of solid oxide fuel cells continue, such reversible PtG systems will remain economically viable even in the presence of substantially lower hydrogen prices in the future. This is because the inherent flexibility in these systems enables them to respond to lower hydrogen prices by operating more frequently in reverse mode, delivering additional electricity to the power markets.

## Results

### Real-time operation of reversible Power-to-Gas.
We examine reversible PtG systems that can (i) produce hydrogen via water electrolysis and (ii) produce electricity from hydrogen and oxygen[26]. We refer to such systems as modular if the two production processes run on separate equipment, such as a one-directional electrolyzer for hydrogen production and a one-directional fuel cell or gas turbine for the reverse operation. In contrast, we refer to a unitized regenerative fuel cell based on, for instance, a SOC[10,27] or a proton exchange membrane (PEM)[22,28] technology as an integrated reversible PtG system. Such systems can carry out both production processes on the same equipment, yet they can only run in at most one direction at any point in time.

Since our interest is in the economics of reversible PtG systems, we focus on such systems operating on their own as price takers in a wholesale market for electricity in which prices are determined hourly based on supply and demand. Time is modeled as a continuous variable $t$ ranging from 0 to 8760 h per year. Let $q(t)$ denote the market price for electricity per kilowatt-hour (kWh) at time $t$. We initially assume that the annual distribution of power prices remains constant across the lifetime of the system. Symbols and acronyms are listed in Supplementary Table 1.

Our model framework considers reversible PtG systems with a peak capacity in kilowatt (kW) of electricity input or output. The assumed size of a PtG system is in line with average capacity sizes reported in the literature[29]. PtG systems generally exhibit economies of scale over a certain range in the sense that system prices per kW, that is, the upfront capital cost decline as the capacity size increases up to a particular level[30]. The numerical calibration of our model relies on parameters for both system prices and operating costs that reflect average system sizes as reported in the existing literature[29].

The basic version of our model makes the simplifying assumption that either reversible PtG system can be brought instantaneously from a cold start to full operating temperature without any loss in conversion efficiencies. Earlier work, however, shows that the process of heating SOCs up to operating temperature can require up to 20 min to prevent excessive material stress[31]. The electrical energy required for that heating amounts to a fraction of the energy needed for the subsequent electrolytic hydrogen production[32]. Reversible PtG systems based on PEM technology can be heated to operating temperature in less than 10 min[30].

We examine the losses incurred by bringing either reversible PtG system from a cold start to full operating temperature in an extension to the basic model provided in Supplementary Note 1. The extension explicitly accounts for (i) the time required to heat either reversible PtG system from a cold start to regular operating temperature, (ii) the energy required for the heating process, (iii) the cost of electricity or hydrogen incurred during the heating period, and (iv) the frequency of those heating periods in each year of operation. On the basis of conservative assumptions for all four of these frictions, our numerical results show that heating costs have only a small impact on the cost of either PtG system. The main reason is that the optimized PtG systems go only through a few heat-up phases per year.

Once the electrolyzer and fuel cell technologies we consider have reached their operating temperature, up- and down-ramping can be conducted in seconds[10,22,30]. The corresponding capacity factors reflect the percentage of the available capacity utilized at time $t$ and can then be chosen flexibly on the interval [0, 1]. Efficiency losses incurred for maintaining the operating temperature are included in the conversion efficiencies considered throughout our analysis. Heat management is commonly more complex for high-temperature electrolyzers and fuel cells, such as SOC facilities, than for low-temperature PEM systems. Nevertheless, the cost of maintaining the operating temperature of well-insulated SOC systems is likely minor[32].

If the modular system generates hydrogen at time $t$, it earns a "conversion price" consisting of the market price of hydrogen, $p$, per kilogram (kg) multiplied with the conversion rate of going from electricity to hydrogen (in kg/kWh). The hydrogen price, $p$, is modeled as time-invariant, because buyers and suppliers typically agree on time-invariant prices[33]. The corresponding conversion parameter $\eta_h^o(CF_h^o)$ represents the amount of hydrogen (in kg) that can be generated from 1 kWh of electricity, given the capacity factor of $CF_h^o$ at time $t$, with $0 \leq CF_h^o \leq 1$. The variable cost of hydrogen generation equals $q(t)$ plus a cost increment $w_h^o$ per kWh that accounts for consumable inputs, like water and reactants for deionizing the water, as well as any purchasing markups on the wholesale price of electricity.

Given a hydrogen price, $p$, the contribution margin per kWh from hydrogen production with the modular reversible PtG system at time $t$ thus is:

$$CM_h^o(CF_h^o, t|p) = [\eta_h^o(CF_h^o) \cdot p - q(t) - w_h^o] \cdot CF_h^o, \quad (1)$$

with $CF_h^o$ to be chosen at each point in time $t$.

Conversely, if the modular system generates electricity, it earns $q(t)$ and incurs a variable cost that comprises $p$ and an incremental cost, $w_e^o$, per kWh of electricity for transporting hydrogen to the Gas-to-Power (GtP) system. To account for efficiency losses, the cost of hydrogen, $p$, is marked-up by the conversion rate for power generation, $\eta_e^o(CF_e^o)$ (in kWh/kg). The shape of the functions $\eta_h^o(\cdot)$ and $\eta_e^o(\cdot)$ depends on the particular technology considered. The contribution margin of electricity generation per kWh at time $t$ then becomes:

$$CM_e^o(CF_e^o, t|p) = \left[ q(t) - \frac{p}{\eta_e^o(CF_e^o)} - w_e^o \right] \cdot CF_e^o. \quad (2)$$

Efficient utilization of the existing capacity is obtained if the capacity factors are at each point in time chosen to maximize the total available contribution margin. While the modular system can run at full capacity in both directions, the 1st Law of Thermodynamics stipulates that the overall round-trip efficiency must satisfy the inequality $\eta_h^o(\cdot) \cdot \eta_e^o(\cdot) \leq 1$ for all $0 \leq CF_h^o, CF_e^o \leq 1$. Consequently, at most one of the terms $[\eta_h^o(CF_h^o) \cdot p - q(t) - w_h^o]$ or $\left[ q(t) - \frac{p}{\eta_e^o(CF_e^o)} - w_e^o \right]$ will be positive for any given values $w_h^o, w_e^o \geq 0$. As illustrated in Fig. 1 (see Methods for formal derivations), efficient system utilization thus implies that the capacity factors be chosen so that $CF_h^o \cdot CF_e^o = 0$. Specifically, the optimal capacity factors, $CF_h^{o*}(t|p)$ and $CF_e^{o*}(t|p)$ maximize

pointwise the sum of the contribution margins in (1) and (2) (see Methods for details). When aggregated across the hours of a year, the maximized contribution margins will be denoted by $CM_h^o(p)$ and $CM_e^o(p)$.

For the integrated system, the economic trade-off is principally the same, except that the incremental cost and conversion rates may differ and instead assume the values $w_h$, $w_e$, $\eta_h(\cdot)$, and $\eta_e(\cdot)$, respectively. Once they are at operating temperature, unitized regenerative fuel cells based on SOC or PEM technology can rapidly switch between hydrogen and electricity production at full capacity[22,27]. Figure 1 illustrated that provided there are no sudden jumps in electricity prices, time intervals where electricity generation is valuable will typically be followed by a time interval in which the system is idle before entering a stretch of time where the regenerative fuel cell again becomes active in either mode of operation.

By construction, the integrated system faces the technical rather than economic "complementary slackness" condition $CF_h \cdot CF_e = 0$ for all $t$. The corresponding contribution margins are:

$$CM_h(CF_h, t|p) = [\eta_h(CF_h) \cdot p - q(t) - w_h] \cdot CF_h, \quad (3)$$

for hydrogen production, and

$$CM_e(CF_e, t|p) = \left[ q(t) - \frac{p}{\eta_e(CF_e)} - w_e \right] \cdot CF_e, \quad (4)$$

for electricity. The capacity factors that maximize the sum of the contribution margins in (3) and (4), subject to the complementary slackness constraint, are denoted by $CF_h^*(t|p)$ and $CF_e^*(t|p)$, respectively. Given these capacity factors, we denote by $CM(p)$ the optimized aggregate contribution margin which is obtained as the total contribution margin obtained after integrating (3) and (4) across the hours of the year.

**Cost competitiveness and the value of reversibility.** A reversible PtG system is said to be cost-competitive if the required upfront investment in equipment yields a positive net present value in terms of discounted future cash flows. The discounted annual stream of optimized contribution margin of the system must then at least cover the initial equipment expenditure. For direct comparison, it will be convenient to capture this economic trade-off on a levelized basis. Analogous to the commonly known levelized cost of electricity, the levelized fixed cost (LFC) of a reversible PtG system reflects the unit acquisition cost of the

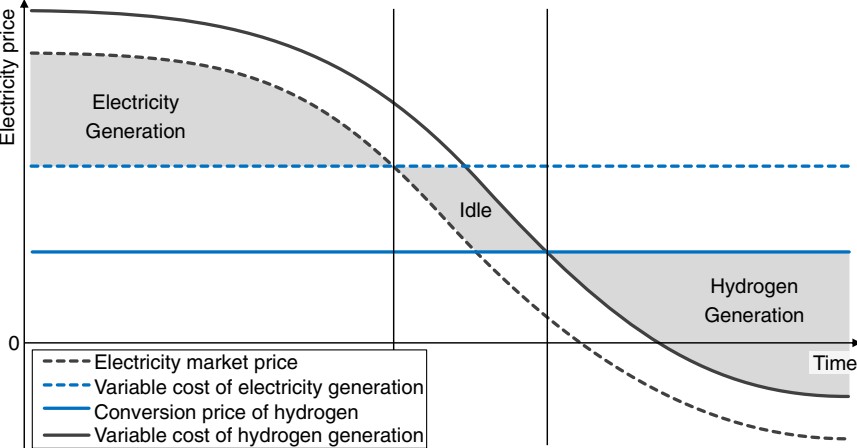

**Fig. 1 Contribution margins of a reversible Power-to-Gas system.** The figure illustrates the three alternative operating modes for either a modular or an integrated reversible PtG system that emerge for varying electricity prices. Wholesale electricity prices can turn negative as a result of surplus energy being supplied to the grid at certain hours.

system per kWh, including applicable fixed operating costs, corporate income taxes, and the cost of debt and equity[34,35].

For the modular system, the LFC for the electrolyzer is denoted by $LFC_h^o$. As shown in Methods, the PtG subsystem is cost-competitive (positive net present value) if and only if at the prevailing market price for hydrogen, $p$:

$$CM_h^o(p) - LFC_h^o > 0. \tag{5}$$

Since the contribution margin from hydrogen is increasing in the selling price of hydrogen, there exists a unique break-even price, $p_h^o$, such that PtG will be cost-competitive whenever $p \geq p_h^o$. Similarly, the Gas-to-Power subsystem is cost-competitive whenever:

$$CM_e^o(p) - LFC_e^o > 0, \tag{6}$$

with $LFC_e^o$ denoting the corresponding LFC per kWh. Since the contribution margin from producing electricity is decreasing in the input price for hydrogen, $p$, there also exists a unique break-even price, $p_e^o$, below which GtP will be cost-competitive.

By design, investors in a modular system retain the option of acquiring only one of the two subsystems. We, therefore, call the modular system cost-competitive if at least one of its subsystems is cost-competitive. In addition, the reversibility feature of the system is said to be valuable if both subsystems have positive net present value on their own. The following finding links cost-competitiveness and the value of reversibility to the prevailing market price of hydrogen.

Finding 1: The modular reversible PtG system is cost-competitive if and only if at the prevailing hydrogen market price, $p$, either $p > p_h^o$ or $p < p_e^o$. Reversibility of the modular system is valuable if and only if $p \in [p_h^o, p_e^o]$.

Figure 2a illustrates the setting of a modular reversible PtG system that is cost-competitive and for which reversibility is valuable. Note that reversibility of the modular system cannot be of value unless $p_h^o < p_e^o$.

For the integrated reversible PtG system, the LFC of the system is denoted by $LFC$. Cost competitiveness of the integrated system then requires that the optimized aggregate contribution margin, $CM(p)$, exceeds $LFC$. The reversibility of the integrated system is said to be valuable if at the prevailing market price of hydrogen, $p$, investment in the system is cost-competitive and, furthermore, the system operates in both directions for select hours of the year, i.e., both sets $\{t|CF_h^*(t|p) > 0\}$ and $\{t|CF_e^*(t|p) > 0\}$ have positive length across the hours of the year.

Figure 2b illustrates a setting in which the reversibility feature of the integrated reversible PtG system is valuable. We note that when viewed as a function of $p$, the optimized contribution margin, $CM(\cdot)$, is drawn as a U-shaped curve. This follows directly from the convexity of this function in $p$ (see Methods), combined with the observation that $CM(p)$ is increasing for large values of $p$ and again increasing as $p$ becomes small, possibly negative. The U-shape of $CM(\cdot)$ implies that there exist at most two break-even points at which $CM(p) = LFC$. These points are denoted by $p_*$ and $p^*$, respectively.

To examine the value of reversibility, suppose hypothetically that the integrated system could operate in only one direction. For instance, suppose the system is constrained to only produce hydrogen (i.e., $CF_e$ in (4) is set identically equal to zero for all $t$). For sufficiently large values of $p$, there then exists a critical value denoted by $\bar{p}$ such that $CM(\bar{p}) = CM_h(\bar{p})$. This equality holds for all $p \geq \bar{p}$. Conversely, there exists a lower critical price below which only electricity generation would be valuable, that is, $CM(p) = CM_e(p)$ for all $p \leq \underline{p}$.

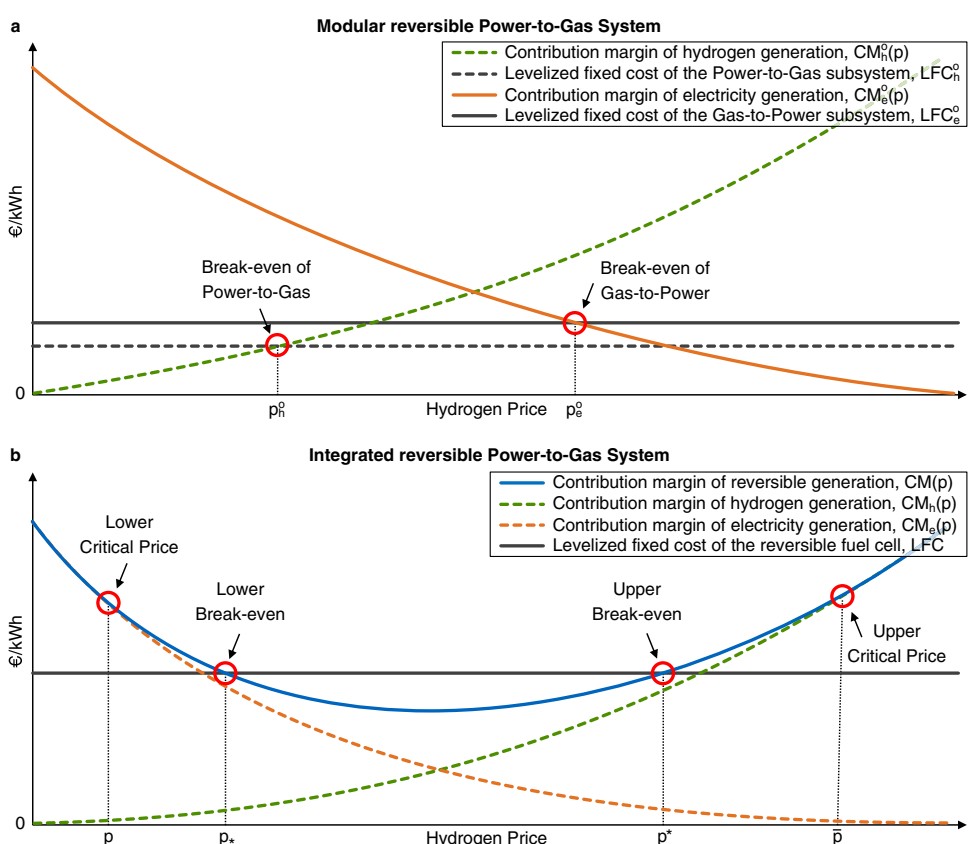

**Fig. 2 Economics of a reversible Power-to-Gas system.** The figure illustrates the potential cost-competitiveness and value of reversible operation in terms of the respective break-even prices of (**a**) a modular reversible Power-to-Gas system, and (**b**) an integrated reversible Power-to-Gas system.

**Finding 2:** The integrated reversible PtG system is cost-competitive if and only if the prevailing hydrogen market price, $p$, does not fall into the range $[p_*, p^*]$. Reversibility of the integrated system is valuable if and only if either $p \in (\underline{p}, p_*)$ or $p \in (p^*, \bar{p})$.

Finding 2 shows that an integrated reversible PtG system is cost-competitive if the market price of hydrogen moves either into an upper or lower range relative to the price at which the optimized contribution margin reaches its minimum. For the case where $p \in (p^*, \bar{p})$, Fig. 2b depicts the possibility that the integrated system primarily generates hydrogen but also operates bi-directionally. Such systems could create an effective buffer against the intermittency of renewables when power is absorbed from the electricity market for hydrogen conversion, yet occasionally electricity is generated at hours of limited power supply and correspondingly high power prices. The range of hydrogen prices at which an integrated system generates both outputs hinges, in addition to cost, on the round-trip efficiency and the volatility in power prices (Fig. 1).

An implicit assumption underlying Finding 2 and Fig. 2b is that LFC exceeds the minimum of the $CM(\cdot)$ curve, for otherwise the break-even prices $p_*$ and $p^*$ do not exist (we ignore the non-generic scenario in which there is exactly one break-even price at a tangential point). In case $LFC < CM(\cdot)$ for all $p$, the integrated reversible PtG system will always be cost-competitive and reversibility will be of value for all hydrogen prices within the interval $(\underline{p}, \bar{p})$. In this case, the flexibility of the integrated reversible PtG system allows it to compensate for any decline in the prevailing market price of hydrogen by turning to electricity production for a larger share of the available time.

**Current economics of reversible Power-to-Gas.** To apply the preceding model framework, we calibrate the model parameters in the current market environment of Germany and Texas. Both jurisdictions have recently deployed considerable amounts of renewable energy[36]. While Germany has maintained coal and natural gas plants as capacity reserves, Texas has retired several conventional generators[37]. The average wholesale electricity price in 2019 was comparable for both jurisdictions, yet power prices in Texas exhibited much higher volatility. As detailed further in Methods and Supplementary Tables 2–5, our calculations rely on a range of data sources collected from journal articles, industry data, and publicly available reports. Table 1 summarizes the average values of key cost and operational parameter estimates.

Our numbers for the modular PtG system are based on a one-directional PEM electrolyzer and a combined-cycle gas turbine.

Recent literature attributes about the same conversion rate to stationary PEM fuel cells as to combined-cycle gas turbines, though the former also entail higher system prices[20,38]. For the integrated reversible PtG system, we consider unitized generative SOC fuel cells that are already commercially available[30,38]. Regarding the conversion efficiency, we note that PEM electrolyzers attain a near-constant efficiency beyond a small threshold utilization level[30]. For integrated PtG systems, we interpret the conversion efficiency parameters identified in the literature (shown in Table 1) as those obtained at full capacity utilization. Thus far, the existing literature provides no evidence on the shape of the functions $\eta_h(\cdot)$ and $\eta_e(\cdot)$. If these conversion rates were to decrease significantly for capacity utilization values approaching one, our findings regarding the cost competitiveness of integrated reversible PtG systems should be interpreted as a lower bound, because the achievable optimized contribution margins might then be larger for capacity factors strictly less than one. Supplementary Note 2 further examines the sensitivity of our numerical findings to changes in the conversion rates of such systems.

The investing party is assumed to have access to the day-ahead wholesale market for electricity (see Supplementary Note 3 for findings based on the real-time wholesale market for electricity). In order to accelerate the transition towards renewable energy, the German government recently decided that electricity purchases for water electrolysis are exempted from certain taxes and fees paid by industrial customers[39]. In Texas, the investing party is assumed to be able to purchase electricity at wholesale prices subject to a markup, as imposed by suppliers like Griddy (see Supplementary Tables 4–5).

We first determine the hydrogen break-even prices. To assess the cost competitiveness of each (sub-)system, we then compare the break-even prices to prevailing transaction prices for hydrogen supply. These values are applicable benchmarks for hydrogen as both an input and an output when the PtG (or GtP) system can be installed nearby a hydrogen or electricity customer. Market prices currently fall into three segments that vary with purity and scale (volume): large-scale supply between 1.5 and 2.5 €/kg, medium-scale between 3.0 and 4.0 €/kg, and small-scale above 4.0 €/kg[33].

Our calculations yield break-even prices for the modular electrolyzer ($p_h^o$) of 3.18 €/kg in Germany and 2.98 \$/kg in Texas, while the break-even prices for the modular gas turbine ($p_e^o$) are 0.57 €/kg in Germany and 1.31 \$/kg in Texas (Table 2, see Supplementary Tables 6–7 for details). The much higher break-

---

**Table 1 Main input variables.**

|  | Germany | Texas |
|---|---|---|
| *Modular reversible PtG system* |  |  |
| Electrolysis: System price | 1606 €/kW | 1799 \$/kW |
| Electrolysis: Conversion rate to hydrogen | 0.019 kg/kWh | 0.019 kg/kWh |
| Gas Turbine: System price | 1000 €/kW | 1199 \$/kW |
| Gas Turbine: Conversion rate to electricity | 20.00 kWh/kg | 20.00 kWh/kg |
| Useful lifetime | 25 years | 25 years |
| *Integrated reversible PtG system* |  |  |
| System price | 2243 €/kW | 2512 \$/kW |
| Conversion rate to hydrogen | 0.023 kg/kWh | 0.023 kg/kWh |
| Conversion rate to electricity | 20.00 kWh/kg | 20.00 kWh/kg |
| Useful lifetime | 15 years | 15 years |
| *Either system* |  |  |
| Average electricity price (2019) | 3.77 €¢/kWh | 3.77 \$¢/kWh |
| Cost of capital | 4.00% | 6.00% |

Conversion rates are based on original industry data and reflect system-level energy efficiencies that include the energy required for maintaining operating temperature. System prices and operating costs reflect average system sizes as reported in the literature[29].

| Table 2 Current economics. | | |
| --- | --- | --- |
| | **Germany** | **Texas** |
| *Modular reversible PtG system* | | |
| Break-even price of Power-to-Gas: $p_h^o$ | 3.18 €/kg | 2.98 $/kg |
| Break-even price of Gas-to-Power: $p_e^o$ | 0.57 €/kg | 1.31 $/kg |
| *Integrated reversible PtG system* | | |
| Upper break-even price: $p^*$ | 3.38 €/kg | 2.78 $/kg |
| Lower break-even price: $p_*$ | 0.03 €/kg | −0.09 $/kg |
| Upper critical price: $\bar{p}$ | 2.43 €/kg | > 5.0 $/kg |
| Lower critical price: $\underline{p}$ | −1.81 €/kg | 0.59 $/kg |

even price for the GtP system in Texas must be attributed to the higher volatility in Texas wholesale electricity prices, which in 2019 exceeded 0.15 $¢/kWh on a regular basis.

Finding 1 implies that modular PtG conversion is cost-competitive in both jurisdictions relative to the prices paid for small- and medium-scale hydrogen supply, while the GtP subsystem is not. Furthermore, the reversibility of the modular system cannot be valuable relative to any prevailing market price for hydrogen because $p_h^o > p_e^o$ in both jurisdictions. Our results here confirm the commonly held view that one-directional GtP systems are currently not economically viable[9,19,20].

For the integrated system, our calculations yield break-even prices of 0.03 €/kg for $p_*$ and 3.38 €/kg for $p^*$ in Germany, while the break-even prices in Texas are −0.09 $/kg and 2.78 $/kg, respectively (Table 2). The substantially smaller $p^*$ in Texas reflects the higher volatility in wholesale power prices. By Finding 2, the integrated system is thus cost-competitive when hydrogen is sold to small- and medium-scale customers in Germany. In Texas, cost competitiveness is achieved even relative to a hydrogen price of at least $2.78 per kg, a value that is borderline for industrial-scale supply.

Regarding the value of reversibility for the integrated system, our calculations yield upper and lower critical prices ($\underline{p}$ and $\bar{p}$) of −1.81 €/kg and 2.43 €/kg, respectively, in Germany. In Texas, the corresponding values are 0.59 $/kg for $\underline{p}$, while $\bar{p}$ exceeds 5.0 $/kg. Because the hydrogen prices for medium scale supply fall "comfortably" into the range $(p^*, \bar{p}) = (2.78, 5.0)$, we conclude that the reversibility of the integrated PtG system is already valuable in the current Texas environment. Contrary to frequently articulated views in the popular press, the generation of electric power from hydrogen is therefore already economical, provided such generation is part of an integrated PtG system that mainly produces hydrogen yet only occasionally operates in the reverse direction to generate electricity. Such systems can therefore be effective in buffering the increasing volatility in power markets resulting from the growing reliance on intermittent renewable energy sources.

A direct comparison of the modular one-sided and the integrated reversible PtG systems shows that the latter is already positioned more competitively despite its substantially higher systems price, as the break-even price of $2.78 per kg is below the corresponding $2.98 per kg for the modular electrolyzer.

**Prospects for reversible Power-to-Gas.** Recent technological progress in reversible PtG systems suggests further improvements in terms of declining system prices and increasing conversion efficiencies[40–42]. System prices of PEM electrolyzers are forecast to decline at an annual rate of 4.77%, while conversion rates are likely to increase linearly to on average 0.023 kg/kWh by 2030[20,33,43]. For combined-cycle gas turbines, both of these parameters are expected to remain unchanged.

To assess the cost dynamics of the unitized generative SOC fuel cell, we rely on a hand-collected data set of $N = 79$ price observations, as described in Methods. We estimate the trajectory of system price by means of a univariate regression covering the years 2000–2019. The functional form of the regression is a constant elasticity model of the form: $v(i) = v(0) \cdot \beta^i$, with $v(i)$ representing the system price in year $i$. As shown in Fig. 3, the resulting estimate for the annual price decline is 8.95% ($\beta = 0.9105$) with a 95% confidence interval of ± 3.20% ($R^2 = 0.27$).

The conversion rate of the regenerative SOC fuel cell is expected to increase linearly to on average 0.024 kg/kWh for hydrogen generation and 21.67 kWh/kg for power generation by 2030[20,38]. Our calculations are based on the current distribution of power prices to isolate the effects of falling system prices and improved conversion rates. A fall in the average of power prices in connection with rising price volatility, as previous studies suggest[13,14,44,45], would affect the economics of either system favorably.

Our model results in a trajectory of break-even prices through 2030 as shown in Fig. 4 (see Supplementary Tables 8, 9 for details). The green lines indicate that the modular electrolyzer is likely to become cost-competitive even relative to the lower prices in the large-scale hydrogen market segment. This conclusion emerges sooner in Texas due to higher volatility in power prices. The break-even prices for the modular gas turbine, as depicted by the orange lines, are projected to remain unchanged. Even though the gap between $p_h^o$ and $p_e^o$ is shrinking, the reversibility feature of the modular system is unlikely to become valuable during the next decade. This stands in contrast to recent ambitions by gas turbine equipment manufacturers[46–48].

The integrated system, in contrast, is projected to become widely cost-competitive for large-scale hydrogen supply in both jurisdictions as shown by the upper blue lines in Fig. 4. We furthermore project the reversibility feature of the integrated system to become increasingly valuable in both jurisdictions as indicated by the falling upper blue lines. In fact, for Texas, the range $[p_*, p^*]$ is almost closing by the end of the coming decade. As explained in the modeling section, a closing of the range corresponds to the scenario where the flexibility inherent in the unitized regenerative fuel cell allows it to achieve an optimized contribution margin that exceeds the LFC of the system, regardless of the prevailing hydrogen price.

In Germany, the reversibility feature of the integrated system is likely to deliver value starting in the second half of the coming decade. This can be seen in Fig. 4a by comparing the upper blue line with the blue dots, which illustrate the trajectory of the upper critical prices ($\bar{p}$) for the integrated system. The reason is that, as the upper break-even price falls, the reversible PtG system will increasingly switch to power generation, as opposed to staying idle, when electricity prices peak (Fig. 1).

**Discussion**
Our analysis has demonstrated that recent advances in unitized regenerative solid oxide fuel cells already make such systems competitive relative to current hydrogen prices. By taking advantage of fluctuations in hourly electricity prices, reversible PtG systems not only act as buffers in electricity markets, they also broaden the supply sources for hydrogen as an industrial input and general energy carrier. If recent trends in the acquisition cost of SOCs continue over the next 5–10 years, our projections indicate that such systems will remain competitive even in the face of substantially lower hydrogen prices, as the electrolyzer then adjusts by operating more frequently as a Gas-to-Power system.

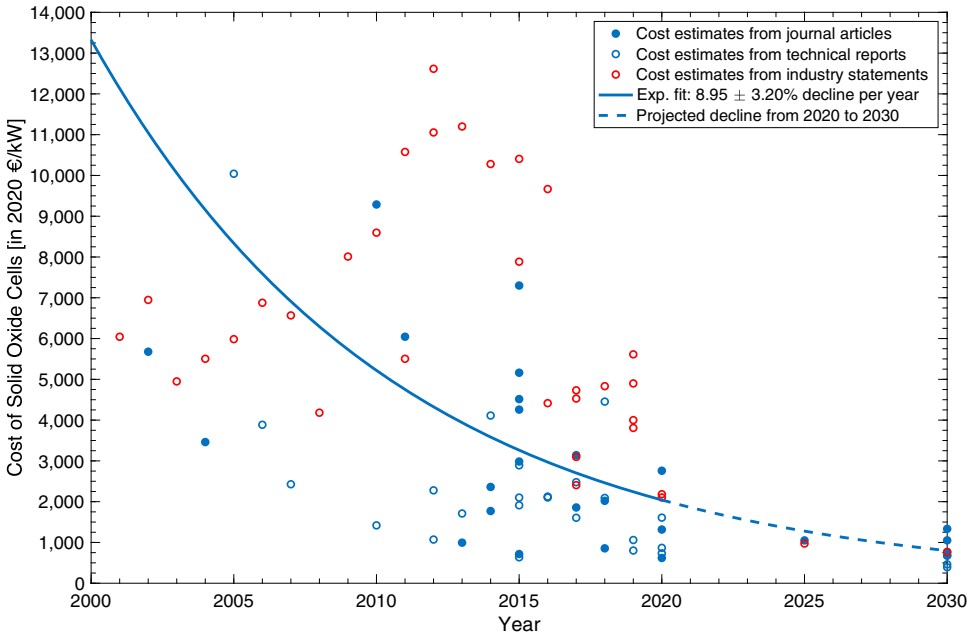

**Fig. 3 Cost of solid oxide cells.** Cost data comes from multiple sources. The univariate regression suggests a constant cost decline over the coming years. The fairly large variance in system prices illustrates the relative novelty of the technology.

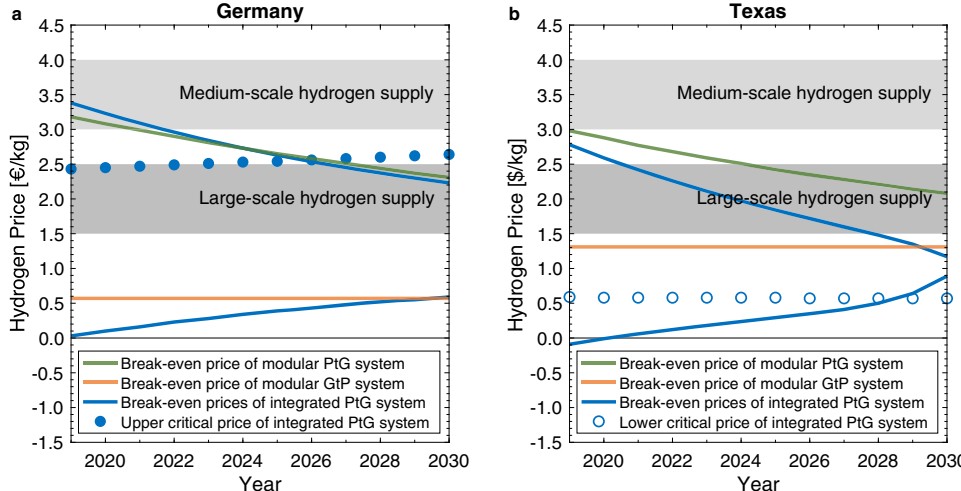

**Fig. 4 Trajectory of break-even and critical hydrogen prices.** This figure contrasts the relevant hydrogen prices of modular and integrated reversible Power-to-Gas systems in (**a**) Germany and (**b**) Texas with the hydrogen prices attained in different market segments. The lower critical price of the integrated system in Germany is consistently below −1.5 €/kg. The upper critical price of the integrated system in Texas is consistently above 5.0 $/kg.

Several promising avenues for future research emerge from our analysis. Earlier work has shown that the economics of electrolyzers can be improved by vertically integrating them with upstream renewable energy sources to achieve operational synergies[49]. It remains to be seen to what extent the addition of a renewable power source would improve the capacity utilization of a reversible PtG system and, therefore, lower the corresponding break-even values. Furthermore, if one views a reversible PtG system as an energy storage device, the natural question is how its competitiveness compares to that of other storage technologies, such as batteries or pumped hydro-power systems[8,9,50,51].

From an industry and policy perspective, we note that the inherent flexibility of integrated reversible PtG systems makes them valuable during periods of electricity scarcity, including regular demand peaks and irregular supply shocks. With increasing penetration levels of renewable energy, this flexibility feature is likely to become more valuable over time. We finally note that our projections regarding the economic positioning of reversible PtG systems in the future have relied on a regression model that presumes that cost declines are a function of calendar time. Yet, the literature on clean energy technologies has shown that cost declines are not merely an exogenous function of time but instead are determined endogenously by the cumulative number of deployments of these systems[43]. Policy-makers should keep these long-term benefits in mind in adopting regulatory policies aimed at accelerating the rate of PtG system deployments in the short run.

## Methods
**Derivation of the aggregate contribution margins.** We begin with the derivation of the optimized aggregate contribution margin, $CM(p)$, that is attainable annually if the investor acquires a 1 kW system of the integrated reversible PtG system and

the prevailing market price of hydrogen is $p$. By construction:

$$CM(p) = \frac{1}{m} \int_0^m \max_{CF_h, CF_e} \left\{ [\eta_h(CF_h) \cdot p - q(t) - w_h] \cdot CF_h \right.$$
$$\left. + [q(t) - \frac{p}{\eta_e(CF_e)} - w_e] \cdot CF_e \right\} dt, \quad (7)$$

subject to $0 \le CF_h, CF_e \le 1$ and the technical constraint that the unitized regenerative fuel cell can only run in one direction at any point in time. It follows that $CM(p)$ is additively separable and can be written as $CM(p) = CM_h(p) + CM_e(p)$, with:

$$CM_h(p) = \frac{1}{m} \int_0^m [\eta_h(CF_h^*(t|p)) \cdot p - q(t) - w_h] \cdot CF_h^*(t|p) dt,$$
$$CM_e(p) = \frac{1}{m} \int_0^m \left[ q(t) - \frac{p}{\eta_e(CF_e^*(t|p))} - w_e \right] \cdot CF_e^*(t|p) dt. \quad (8)$$

Here, $CF_h^*(t|p)$ and $CF_e^*(t|p)$ are chosen to maximize $[\eta_h(CF_h) \cdot p - q(t) - w_h]$ and $[q(t) - \frac{p}{\eta_e(CF_e)} - w_e]$, respectively, at all points in time $t$.

For the modular reversible PtG systems, the aggregate optimized contribution margins $CM_h^o(p)$ and $CM_e^o(p)$ are derived in direct analogy to (8).

**Convexity of $CM(\cdot)$ in $p$.** We demonstrate the convexity of the aggregate annual contribution margin pointwise, that is, convexity holds at any point in time $t$. Specifically, it suffices to show that for any $0 \le \lambda \le 1$:

$$CM(t|p_\lambda) = A(t|p_\lambda) \cdot CF_h^*(t|p_\lambda) + B(t|p_\lambda) \cdot CF_e^*(t|p_\lambda)$$
$$\le \lambda \left[ A(t|p^1) \cdot CF_h^*(t|p^1) + B(t|p^1) \cdot CF_e^*(t|p^1) \right]$$
$$+ (1-\lambda) \left[ A(t|p^0) \cdot CF_h^*(t|p^0) + B(t|p^0) \cdot CF_e^*(t|p^0) \right] \quad (9)$$
$$= \lambda \cdot CM(t|p^1) + (1-\lambda) \cdot CM(t|p^0),$$

where $p_\lambda \equiv \lambda \cdot p^1 + (1-\lambda) \cdot p^0$, $A(t|p) \equiv \eta_h(CF_h^*(t|p)) \cdot p - q(t) - w_h$ and $B(t|p) \equiv q(t) - \frac{p}{\eta_e(CF_e^*(t|p))} - w_e$. As noted above, for any $p$, either $A(t|p) \le 0$ or $B(t|p) \le 0$ because $\eta_h(\cdot) \cdot \eta_e(\cdot) \le 1$.

Suppose now, without loss of generality, that $A(t|p_\lambda) > 0$ in which case the left-hand side of the preceding inequality is equal to $A(t|p_\lambda)$. Finally, the right-hand side of the above inequality is given by:

$$\lambda \cdot \max\{A(t|p^1), B(t|p^1), 0\} + (1-\lambda) \cdot \max\{A(t|p^0), B(t|p^0), 0\}. \quad (10)$$

By construction, this last expression is at least as large as $\lambda \cdot A(t|p^1) + (1-\lambda) \cdot A(t|p^0)$, which, because of the linearity of $A(t|\cdot)$ in $p$, is equal to $A(t|p_\lambda)$, thus establishing the desired inequality. The claim regarding convexity of $CM(\cdot)$ then follows from the observation that the sum (integral) of convex functions is also convex.

**Net present value of the reversible PtG systems.** As before, we focus on integrated reversible PtG systems, with the derivation for modular systems being entirely analogous. The $LFC$ of the system, $LFC$, aggregates all fixed expenditures required over the life of the reversible PtG system. This aggregate expenditure is then divided by $L$, the levelization factor that expresses the discounted number of hours that the capacity is available over its lifetime. The resulting cost is then a unit cost per hour of operation. Formally:

$$LFC = f + \Delta \cdot c. \quad (11)$$

Here, $f$ represents the levelized value of fixed operating costs, $c$ represents the levelized capacity cost per kWh, and $\Delta$ captures the impact of income taxes and the depreciation tax shield. Denoting by $v$ the system price of the regenerative fuel cell per kW of peak electricity absorption and desorption, we have:

$$c = \frac{v}{L}, \quad (12)$$

with the levelization factor calculated as:

$$L = m \cdot \sum_{i=1}^T \gamma^i \cdot x^{i-1}. \quad (13)$$

Here, $m$ denotes the number of hours per year, that is, $m = 8760$ and the parameter $T$ represents the number of years of useful economic life of the system. Since capacity may degrade over time, we denote by $x$ the degradation factor so that $x^{i-1}$ gives the fraction of the initial capacity that is functioning in year $i$. The parameter $\gamma = (1+r)^{-1}$ and represents the discount factor with $r$ as the cost of capital. This interest rate should be interpreted as the weighted average cost of capital if the levelized cost is to incorporate returns for both equity and debt investors. Similarly, the levelized fixed operating cost per kWh similarly comprises the total discounted fixed operating cost incurred over the lifetime of the system:

$$f = \frac{\sum_{i=1}^T F_i \cdot \gamma^i}{L}. \quad (14)$$

The cost of capacity is affected by corporate income taxes through a debt and a depreciation tax shield, as interest payments on debt and depreciation charges reduce the taxable earnings of a firm. The debt tax shield is included in the calculation if $r$ is interpreted as the weighted average cost of capital. Let $d_i$ denote the allowable tax depreciation charge in year $i$. Since the assumed lifetime for tax purposes is usually shorter than the actual economic lifetime, we set $d_i = 0$ in those years. If $\alpha$ represents the effective corporate income tax rate, the tax factor is given by:

$$\Delta = \frac{1 - \alpha \cdot \sum_{i=0}^T d_i \cdot \gamma^i}{1 - \alpha}. \quad (15)$$

The formal claim then is that the net present value of an investment in one kW of the integrated reversible PtG system is given by:

$$NPV(p) = (1 - \alpha) \cdot L \cdot [CM(p) - LFC]. \quad (16)$$

This relation is readily verified by noting that the after-tax cash flows in year $i$ is:

$$CFL_i(p) = x^{i-1} \cdot \int_0^m CM(t|p) \, dt - F_i - \alpha \cdot I_i(p), \quad (17)$$

where $I_i(p)$ denotes the taxable income in year $i$. Specifically:

$$I_i(p) = x^{i-1} \cdot \int_0^m CM(t|p) \, dt - F_i - v \cdot d_i. \quad (18)$$

Since $CFL_0 = -v$, the discounted value of all after-tax cash flows is indeed equal to the expression in (16). Similar reasoning yields that the unit net present values of the modular PtG and GtP systems are, respectively, given by:

$$NPV_h(p) = (1 - \alpha) \cdot L \cdot \left[ CM_h^o(p) - LFC_h^o \right], \quad (19)$$

and

$$NPV_e(p) = (1 - \alpha) \cdot L \cdot \left[ CM_e^o(p) - LFC_e^o \right]. \quad (20)$$

**Cost dynamics of solid oxide cells.** We collected cost estimates from a range of information sources, including industry publications, academic articles in peer-reviewed journals and technical reports by agencies, consultancies, and analysts. These documents were retrieved by searching the database Scopus and the web with Google's search engine using a combination of one of the five technology-specific keywords 'reversible electrolyzer', 'reversible fuel cell', 'solid oxide electrolysis cell', 'solid oxide fuel cell', or 'reversible PtG' with the two economic keywords 'cost' and 'investment'. For industry statements, we also searched with the name of a manufacturer in combination with the economic keywords. For the Google search, we reviewed the top 100 search results. The review and the data set is documented in an Excel file available as Supplementary Data 1.

The review yielded 211 sources, which we filtered by several criteria to ensure quality and timeliness. First, we excluded results published before the year 2000 and, for journal articles, results published in a journal with a rank below 0.5 in the Scimago Journal and Country Rank. The threshold of 0.5 showed to be effective for excluding articles published, for instance, in conference proceedings without peer-review. As for technical reports, we only included results that could convince through appearance, writing, clarity of methodology, and reputation of the author(s) or authoring organization(s). These measures removed 47 sources. Reviewing the resulting stock of sources, we further excluded sources that did not provide direct cost or efficiency data (49) and sources citing other articles as original sources (29). These citations were traced back to the original source. If the original was new, we added it to the pool. We further added sources that we found with a previous review[33] and that were new to the pool.

Our procedure left 86 sources with original data from industry or an original review of multiple sources and yielded 89 cost estimates. In case the sources issued range estimates, we took the arithmetic mean of the highest and the lowest value. The common currency is Euro and all data points in other currencies were converted using the average exchange rate of the respective year as provided by the European Central Bank. Regarding inflation, all historic cost estimates were adjusted using the HCPI of the Euro Zone as provided by the European Central Bank. The cost estimates were winsorized at a 1.0% level. Figure 3 in the main body shows the cost estimates and regression results.

## Data availability
The data used in this study are referenced in the main body of the paper and the Supplementary Information. Data that generated the plots in the paper are provided in the Supplementary Information. Additional information is available from the corresponding author upon request.

## Code availability
Computational code is available upon request to the corresponding author.

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

## Acknowledgements

We gratefully acknowledge financial support through the Deutsche Forschungsgemeinschaft (DFG, German Research Foundation): Project-ID 403041268, TRR 266. This research was also supported by the Joachim Herz Stiftung and the Hanns-Seidel-Stiftung with funds from the Federal Ministry of Education and Research of Germany. Helpful comments were provided by Stefanie Burgahn, Céleste Chevalier, Stephen Comello, Gunther Friedl, Rebecca Meier, Christian Stoll, Nikolas Wölfing, and colleagues at the University of Mannheim, the Technical University of Munich, and the Massachusetts Institute of Technology. Finally, we thank Lisa Fuhrmann for providing valuable assistance with the data collection.

## Author contributions

G.G. initiated the research question and the techno-economic model. He also conducted the literature review, the data collection, and the numerical calculations. G.G. and S.R. jointly analyzed the economic model and both contributed to the writing of the paper.

## Funding

## Competing interests

The authors declare no competing interests.

**Additional information**

