## [Peer Review File · Nature Communications]

Reversible Power-to-Gas systems for energy conversion and storageReviewers' comments:

Reviewer #1 (Remarks to the Author):

Although the article contains some insightful information on the potential economical value of reversible power-to-H₂ (PtH) systems under specific assumptions, it lacks a solid understanding of the operational details of reversible solid oxide electrolyser and fuel cell technology. For this reason, it cannot be accepted for publication in a high impact journal like Nature Communications.

Fundamental knowledge about the central reversibility aspect is lacking at 3 major levels :

1) (p.2) "The electrolyzer and fuel cell technologies we consider in the empirical part allow for rapid up- or down-ramping [28]. The corresponding capacity factors, i.e. the percentage of the available capacity utilized at time t , can thus be chosen anywhere on the interval $[0; 1]$. We denote these capacity factors by $CF(t)$, and note that, in the context of our model, optimal utilization will always entail a "bang-bang" type solution so that $CF(t)$ is always equal to zero or one."

As a matter of fact, among the 3 main types of water electrolyzer technologies (alkaline, PEM and SOEC), SOEC is the one which is the least (or even not all) suitable for intermittent operation as a direct result of the significant operational cost of heat for maintaining or ramping up temperature (upto 800°C). This cost was not at all accounted for when switching CF between 0 and 1.

Moreover, the cited reference [28] that was used to claim rapid up- or down-ramping does not contain any meaningful information at all about this claim.

2) Rather than considering a "bang-bang" type solution, with a CF being either 0 or 1, the authors should have considered a variable operational working point (which then in turn also leads to a variable conversion rate. They can consult for instance Renewable and Sustainable Energy Reviews 15 (2011) 1–23, where it is explained in detail how the time variability of the cell operating current is one of the most important parameters in terms of its potential impact on the capital cost.

Depending on the intermittency, the electricity price and the investment cost, it is the current-voltage operating point that needs to be varied during the cell lifetime to optimize for the lowest H₂ production cost.

3) One of their previous publications (Nature Energy 4 (2019) 216-222) used a very similar economical approach but then for PEM electrolyser systems under one-directional operation (one may regret that significant parts of the current paper are a mere copy of the used methodology). Contrary to this previous work, it is not clear why there is no consideration whatsoever about the optimal capacity size (power scale) of modular or integrated reversible PtH systems. In line with this comment, it can be noted as well that the relatively large variability in the collected CAPEX data (€/kW) in Fig. 3 is a direct result of their underlying variability on capacity, a fundamental consideration which was not at all included in the current analysis.

Finally, a rather fundamental question can be raised with respect to the starting motivation of the work, namely that (Abstract) "wholesale power markets have seen increasing volatility with significant amounts of surplus electricity at select hours of the year." As a matter of fact, the number of hours per year with negative electricity prices is very minor (cfr. review attachment) .

Reviewer #2 (Remarks to the Author):

The topic is interesting, and the authors have performed a good analysis and investigations; however, I recommend improving the quality of this report by considering the following comments:

I believe it is better to use real-time electricity market price from real data or forecasted data using ML methods.

Did the authors investigate efficiency losses in experiments that will be used in conversion rate for power generation?

I do not agree with the constant cost of the conversion price of hydrogen. The authors should elaborate the method used for defining the prices.

What about the real-time condition of the power output of renewable energy sources and the uncertain nature of PV and wind power output?

Figure 3 is of great importance that shows the cost of solid oxide cells. Cost data comes from multiple sources. The univariate regression suggests a constant cost decline over the coming years. The projection for the future has made it interesting. Is there any analysis that defines the contribution of the PtG systems considering the high penetration rate of renewable energy sources in the future?

**Authors' Response to Reviewers on
"Energy Conversion and Storage: The Value of Reversible Power-to-Gas Systems"
Ms. Ref. No.: NCOMMS-21-06673A-Z, Nature Communications**

We are grateful to the two reviewers for their many constructive comments and suggestions. In response, we have undertaken a comprehensive revision of the manuscript. In particular, we have improved the Abstract, Introduction, and Conclusion to (i) clarify the motivation and scope of our analysis, and (ii) highlight the contributions of our work. We have also substantially revised the model sections to allow for the possibility of conversion efficiencies that vary with the degree of capacity utilization. Finally, we provide an extension to our analysis to explicitly examine the cost of heating reversible PtG systems to operating temperature. We believe the comprehensive revision of the manuscript has resulted in a stronger and more transparent paper.

The following points respond to each reviewer's comments in detail. For that purpose, the original reviewer comments have been italicized, while our responses are shown in regular font.

Response to Reviewer 1

Although the article contains some insightful information on the potential economical value of reversible power-to-H₂ (PtH) systems under specific assumptions, it lacks a solid understanding of the operational details of reversible solid oxide electrolyser and fuel cell technology. For this reason, it cannot be accepted for publication in a high impact journal like Nature Communications.

Fundamental knowledge about the central reversibility aspect is lacking at 3 major levels :

1) (p.2) *"The electrolyzer and fuel cell technologies we consider in the empirical part allow for rapid up- or down-ramping [28]. The corresponding capacity factors, i.e. the percentage of the available capacity utilized at time t , can thus be chosen anywhere on the interval $[0; 1]$. We denote these capacity factors by $CF(t)$, and note that, in the context of our model, optimal utilization will always entail a "bang-bang" type solution so that $CF(t)$ is always equal to zero or one."*

As a matter of fact, among the 3 main types of water electrolyzer technologies (alkaline, PEM and SOEC), SOEC is the one which is the least (or even not all) suitable for intermittent operation as a direct result of the significant operational cost of heat for maintaining or ramping up temperature (upto 800°C). This cost was not at all accounted for when switching CF between 0 and 1. Moreover, the cited reference [28] that was used to claim rapid up- or down-ramping does not contain any meaningful information at all about this claim.

We apologize for the misplaced reference [28] and have carefully checked our references. Regarding the dynamic operation of the fuel cells, our reading of the literature is that the most recent technological advances allow unitized regenerative fuel cells based on, for instance, proton exchange membrane (PEM) or solid oxide cell (SOC) technology to rapidly (within seconds) ramp up or down (regardless of the direction of operation), once they are at full operating temperature. To further examine the cost of reaching the operating temperature from a cold start, we provide an extension to our analysis in the new Supplementary Note 1. The calculations show that for the environments we consider, the overall economic effect of bringing the electrolyzer and/or fuel cell occasionally to full operating temperature is minor. This finding reflects that the total number of minutes lost to having the PtG system brought up to temperature again is not significant. We discuss this point and provide corresponding references in the third paragraph of Section 2.

2) *Rather than considering a "bang-bang" type solution, with a CF being either 0 or 1, the authors should have considered a variable operational working point (which then in turn also leads to a variable conversion rate. They can consult for instance Renewable and Sustainable Energy Reviews 15 (2011) 1–23, where it is explained in detail how the time variability of the cell operating current is one of the most important parameters in terms of its potential impact on the capital cost. Depending*

on the intermittency, the electricity price and the investment cost, it is the current-voltage operating point that needs to be varied during the cell lifetime to optimize for the lowest H₂ production cost.

In the revised version of our paper, the analytical model now allows for the possibility that the conversion efficiency is a function of the contemporaneous capacity utilization. In the empirical part of our analysis, we rely on fixed conversion rates as approximated in the literature, which we interpret as those obtained at full capacity utilization. For modular PEM electrolyzers, the literature suggests that they attain a near-constant conversion efficiency beyond a small threshold utilization level. For integrated PtG systems based on SOC technology, we are not aware of prior studies providing clear evidence on the shape of the conversion efficiencies as a function of capacity utilization. If significantly higher conversion efficiencies can be attained for intermediate degrees of capacity utilization, our findings should be interpreted as lower bounds on the economic viability of integrated PtG systems. We discuss this issue in the second paragraph of Section 4.

To further examine the sensitivity of our numerical findings for integrated reversible PtG systems based on SOC technology, the new Supplementary Note 2 provides two kinds of analyses. First, we examine the magnitude of the effect of the conversion efficiencies at full capacity utilization being higher or lower than the values we identified in the literature. Our findings for this analysis show that the central insights identified in the main body of the paper are generally robust to a fairly wide range of conversion efficiencies.

As a second sensitivity analysis, we examine the trade-off that may arise if conversion efficiencies were to increase significantly for capacity utilization rates of less than one. In particular, we examine the increase in both conversion efficiencies that would have to emerge in order for the system operated at a capacity utilization less than 100% to be just as economical as a system operated at full capacity with the conversion rates we associate with 100% utilization. For hypothetically even higher conversion efficiencies at intermediate capacity utilization levels, our calculations suggest that the overall effect on the economic positioning of the system (expressed in terms of the upper break-even price for hydrogen) changes relatively slowly.

3) One of their previous publications (Nature Energy 4 (2019) 216-222) used a very similar economical approach but then for PEM electrolyser systems under one-directional operation (one may regret that significant parts of the current paper are a mere copy of the used methodology). Contrary to this previous work, it is not clear why there is no consideration whatsoever about the optimal capacity size (power scale) of modular or integrated reversible PtH systems. In line with this comment, it can be noted as well that the relatively large variability in the collected CAPEX data (€/kW) in Fig. 3 is a direct result of their underlying variability on capacity, a fundamental consideration which was not at all included in the current analysis.

Since our main interest is in the economics of reversible PtG systems, we focus on such systems operating on their own in the wholesale electricity market. We have clarified this in Section 2. The vertical integration of a reversible PtG system with a co-located source for renewable electricity might improve the competitive position of the reversible PtG system, as we discuss in the conclusion. Yet, the modeling of such a market setting would require substantial increases in complexity without adding major insights on the economics of reversible PtG systems.

With regard to the capacity sizes of both the modular subsystems and the integrated system, we note that, given the maintained assumption of constant returns to scale technology (i.e., capacity costs scale proportionally with capacity size), there is no loss of generality in normalizing the capacity investments of either system to 1 kW of electricity input or output.

In the empirical part of our analysis, we have clarified in a note to Table 1 that cost parameters used in our analysis account for economies of scale exhibited by systems of an average size. Furthermore, we have expanded the caption of Figure 3 to explain that the fairly large variation in system prices partly reflects the relative newness of the technology.

Finally, a rather fundamental question can be raised with respect to the starting motivation of the work, namely that (Abstract) "wholesale power markets have seen increasing volatility with significant amounts of surplus electricity at select hours of the year." As a matter of fact, the number of hours per year with negative electricity prices is very minor (cfr. review attachment.

In response to your comment, we have substantially revised the Abstract, Introduction, and Conclusion to clarify the relevance and scope of our analysis. Specifically, we have clarified our language insofar as "surplus electricity" does not require power prices to be negative but merely to be relatively low.

Response to Reviewer 2

The topic is interesting, and the authors have performed a good analysis and investigations; however, I recommend improving the quality of this report by considering the following comments: I believe it is better to use real-time electricity market price from real data or forecasted data using ML methods.

Thank you for this suggestion. We have run our calculations again using hourly prices for electricity for the real-time wholesale market in Texas. The resulting break-even prices in the current (2019) economic environment do not differ substantially (if at all) from the break-even prices retrieved based on day-ahead market prices. We have added a corresponding statement to paragraph 3 of section 4. We have also provided the results in the new Supplementary Note 3.

Did the authors investigate efficiency losses in experiments that will be used in conversion rate for power generation?

The conversion rates used in the numerical analysis are based on original experiments and industry data. They also reflect system-level energy efficiencies that include energy required for heat management. We have added a clarifying sentence in the notes to Table 1.

I do not agree with the constant cost of the conversion price of hydrogen. The authors should elaborate the method used for defining the prices.

We model the hydrogen price as time-invariant, because buyers and suppliers typically enter into fixed-price contracts for extended periods of time. In contrast, wholesale market prices for electricity fluctuate considerably, arguably because electricity is more expensive to store. We have added a clarifying statement in paragraph 4 of Section 2.

What about the real-time condition of the power output of renewable energy sources and the uncertain nature of PV and wind power output?

In the context of our model, the real-time output variation in renewable energy generation is reflected in the wholesale market prices for electricity. Fluctuations in output, relative to their mean values, at any given point in time should only be of importance for a risk-averse decision maker. When considering the economic viability of integrated PtG systems, we have effectively focused on risk-neutral decision makers, though a risk premium is principally reflected in the cost of capital.

Figure 3 is of great importance that shows the cost of solid oxide cells. Cost data comes from multiple sources. The univariate regression suggests a constant cost decline over the coming years. The projection for the future has made it interesting. Is there any analysis that defines the contribution of the PtG systems considering the high penetration rate of renewable energy sources in the future?

A rapidly growing body of literature studies the economic viability and/or system value of one-directional electrolyzers in the current electricity system and/or potential future energy systems with higher shares of intermittent renewable energy. A few studies also examine the system value of reversible PtG systems focusing exclusively on modular systems (see, for instance, <https://doi.org/10.1038/nclimate3045> or <https://doi.org/10.1038/s41560-021-00796-8>). As our results suggest, however, integrated reversible PtG systems might be more valuable. It would thus be instructive to examine the potential role of integrated systems in energy systems with high levels of intermittent renewable energy.

In conclusion, we again thank the reviewers for their many insightful comments and suggestions. We believe the resulting revision has made the manuscript more accessible and transparent, and sharpened our message. We hope you agree.

Reviewers' comments:

Reviewer #1 (Remarks to the Author):

After carefully considering the authors' responses as well as the revised manuscript as a whole, there are still a number of fundamental concerns remaining that in our opinion do not justify publication in Nature Comm. Indeed, from the 3 major concerns that have been raised initially about the central reversibility aspect, only the 2nd one (related to the consideration of a capacity dependent conversion efficiency) has been appropriately addressed.

For the 1st one, the authors have now simply specified that for the dynamic operation, their initial modeling was considering a full operating temperature. They have now refined their analysis somewhat by also considering a ramping up from a cold start, but only in terms of a decrease in the effective number of operational minutes as a result of such a ramping up, and not as an additional operational cost due the required heat for realising such temperature ramp up. They simply justify their assumption by stating that (p.3) "Earlier studies have shown that the cost of maintaining the temperature of well-insulated systems is likely minor.²⁹" However, this quote from ref. 29 (a new ref. that we have suggested ourselves after the 1st review) is referring to the integrated use of SOC technology for hydrocarbon fuel production. In that case, a significant amount of heat can be recuperated for pre-heating steam (and CO₂) from the down-stream exothermal Fischer-Tropsch synthesis. As a matter of fact, with respect to SOC technology, ref. 29 explicitly states that (p.13, 2nd paragraph) "However, heat management is more complicated for these high temperature cells and can more easily lead to energy losses, as well as higher capital costs due to materials and equipment failures or simply additional system costs that have been underestimated." The appropriate quantitative energy balances can be found in Table 2 of ref. 29.

For the 3th one, the authors have maintained their assumption of a constant return to scale technology, meaning that capacity costs scale proportionally with capacity size. This is fundamentally wrong, since CAPEX values (in €/kW) for electrolysers are well-documented to decrease with capacity (in kW).

Finally, even if the phrase of significant amounts of surplus electricity at select hours of the year does no longer appear in the revised manuscript, associating the notion of surplus electricity no longer to power prices that are negative but "merely to be relatively low" is not a very convincing quantitative argument as long as this critical price level is not predefined.

Reviewer #2 (Remarks to the Author):

The authors have improved the quality of the paper based on reviewers' comments. I have no more comments.

**Authors' Response to Reviewer 1's Report on
"Energy Conversion and Storage: The Value of Reversible Power-to-Gas Systems"
Ms. Ref. No.: NCOMMS-21-06673A-Z, Nature Communications**

We reproduce Reviewer 1's comments in italics, followed by our responses in regular font.

After carefully considering the authors' responses as well as the revised manuscript as a whole, there are still a number of fundamental concerns remaining that in our opinion do not justify publication in Nature Comm. Indeed, from the 3 major concerns that have been raised initially about the central reversibility aspect, only the 2nd one (related to the consideration of a capacity dependent conversion efficiency) has been appropriately addressed.

For the 1st one, the authors have now simply specified that for the dynamic operation, their initial modeling was considering a full operating temperature. They have now refined their analysis somewhat by also considering a ramping up from a cold start, but only in terms of a decrease in the effective number of operational minutes as a result of such a ramping up, and not as an additional operational cost due the required heat for realising such temperature ramp up. They simply justify their assumption by stating that (p.3) "Earlier studies have shown that the cost of maintaining the temperature of well-insulated systems is likely minor.²⁹" However, this quote from ref. 29 (a new ref. that we have suggested ourselves after the 1st review) is referring to the integrated use of SOC technology for hydrocarbon fuel production. In that case, a significant amount of heat can be recuperated for pre-heating steam (and CO₂) from the down-stream exothermal Fischer-Tropsch synthesis. As a matter of fact, with respect to SOC technology, ref. 29 explicitly states that (p.13, 2nd paragraph) "However, heat management is more complicated for these high temperature cells and can more easily lead to energy losses, as well as higher capital costs due to materials and equipment failures or simply additional system costs that have been underestimated." The appropriate quantitative energy balances can be found in Table 2 of ref. 29.

We fully agree with Reviewer 1 and Graves et al. (2011, formerly cited as ref. 29), that, compared to low-temperature electrolyzers, heat management is "more complicated" for high-temperature electrolyzers, such as solid oxide cells. We therefore consider both the cost of heating a reversible Power-to-Gas (PtG) system to operating temperature and the cost of maintaining that temperature.

As discussed on pages 2-3 of the main text, and in further detail in Supplementary Note 1, we examine the operational cost for heating a reversible PtG system. Our analysis includes (i) the time required to heat the system to regular operating temperature from a cold start, (ii) the energy required for this heating process, (iii) the cost of electricity or hydrogen incurred during this heating period, and (iv) the frequency of those heating periods in each year of operation.

Our numerical analysis of the impact of those heating costs employs conservative assumptions, e.g., the assumption that the heating process requires the same energy per hour as the production process thereafter. On this note, Table 2 of Graves et al. (2011) indicates that the energy required for heating a solid oxide electrolyzer to operating temperature is far less than the energy input needed for the subsequent electrolytic hydrogen production, even in the absence of heat energy available from a downstream exothermal production process. Nonetheless, our calculations show that the overall economic effect of bringing the Power-to-Gas systems occasionally to full operating temperature is not significant. This finding simply reflects that the need for cold starts arises only infrequently in the market environments we consider.

To account for the cost of maintaining the operating temperature, the conversion efficiencies considered throughout our analysis include efficiency losses incurred for maintaining that temperature. This is stated explicitly in the first paragraph of page 3 of the submitted manuscript. As such, our analysis does not assume that the PtG system is subsidized by heat energy obtained from a downstream exothermal production process.

For the 3th one, the authors have maintained their assumption of a constant return to scale technology, meaning that capacity costs scale proportionally with capacity size. This is fundamentally wrong, since CAPEX values (in €/kW) for electrolyzers are well-documented to decrease with capacity (in kW).

We agree with the claim that, as capacity size increases electrolyzers will require lower upfront capital costs per kW of capacity, at least up to a certain size. Our analysis does not seek to optimize the size of the system. Instead, we effectively calibrate all cost parameters, system prices and operating costs, as they have been reported in the existing literature for the system sizes that have been built thus far. We now clarify this choice on page 2-3 of the manuscript. We note parenthetically that if larger systems were to benefit from additional economies of scale, our findings on the value of reversible Power-to-Gas systems would become more favorable.

Finally, even if the phrase of significant amounts of surplus electricity at select hours of the year does no longer appear in the revised manuscript, associating the notion of surplus electricity no longer to power prices that are negative but “merely to be relatively low” is not a very convincing quantitative argument as long as this critical price level is not predefined.

There is no “critical price level” for electricity in our analysis. Our language in the Introduction and Conclusion simply refers to the fact that the range of electricity prices must exhibit sufficient variation across the hours of the year (and our analytical model formalizes what exactly this means) in order for the reversibility of reversible PtG system to be valuable. We find that the annual variation of electricity prices in the state of Texas is already sufficient to arrive at this conclusion for integrated systems.

REVIEWERS' COMMENTS

Reviewer #1 (Remarks to the Author):

Probably based on the (much) less critical report of the 2nd reviewer, the authors seem to have pushed their responses to most of my earlier concerns. Although my own intuition as an experimentalist still says something is "not quite" right, it seems that from their rebuttal letter, most of my concerns have been addressed. This holds in particular for the one about the heat requirements for high T electrolysis, and how this might effect the overall energy balance. As a result, I can now accept the newly revised version.